# An underground, wireless, open-source, low-cost system for monitoring oxygen, temperature, and soil moisture

Elad Levintal[1], Yonatan Ganot[1], Gail Taylor[2], Peter Freer-Smith[2,3], Kosana Suvocarev[1], Helen E Dahlke[1]

[1]Department of Land, Air and Water Resources, University of California, Davis, 95616, CA, USA
[2]Department of Plant Sciences, University of California, Davis, 95616, CA, USA
[3]Forest Research, UK

*Correspondence to*: Elad Levintal (elevintal@ucdavis.edu)

**Abstract.** The use of wireless sensor networks to measure soil parameters eliminates the need to remove sensors for field operations, such as tillage, thus allowing long-term measurements without multiple disturbances to soil structure. Wireless sensors also reduce aboveground cables and the risk of undesired equipment damage and potential data loss. However, implementing wireless sensor networks in field studies usually requires advanced and costly engineering knowledge. This study presents a new underground, wireless, open-source, low-cost system for monitoring soil oxygen, temperature, and soil moisture. The process of system design, assembly, programming, deployment, and power management is presented. The system can be left underground for several years without the need for changing the battery. Emphasis was given on modularity so that it can be easily duplicated or changed if needed, and deployed without previous engineering knowledge. Data from this type of system have a wide range of applications, including precision agriculture and high-resolution modelling.

## 1 Introduction

A remaining challenge in vadose zone monitoring is the measurement of soil parameters, such as water content, without the need to remove sensors between field operations, such as tillage, which often causes damage to wires connecting below-ground sensors to above-ground dataloggers. In addition, the standard setting of using cables to connect underground sensors to an aboveground datalogger and power source can change soil structure by causing macro-pores and fractures. The altered structure can potentially cause unwanted experimental artifacts, such as preferential water flow or higher aeration rates. Also, aboveground cables can be subjective to undesired damage resulting from weather events, agricultural machinery, pests, and animals (Hardie and Hoyle, 2019; Vuran et al., 2018). The use of wireless underground sensor networks (WUSNs), instead of cables, can solve these issues (Cardell-Oliver et al., 2019; Zhang et al., 2017).

While aboveground wireless sensor networks (WSNs) have been readily studied and implemented (DeBell et al., 2019; García et al., 2021), the use of WUSNs is still in its early stages (Hardie and Hoyle, 2019; Wan et al., 2017). A WUSN is defined as a system in which all sensors and communication components are buried underground (node) while wirelessly transmitting

data through the soil to an aboveground hub (also referred to in the literature as a gateway) (Huang et al., 2020). The definition of an underground node is study-dependent, here a node is defined as a single transmitting system located at a single depth; different sensors can be connected to one node via underground cables. The most basic WUSN configuration includes a single underground node and an aboveground hub. Advanced WUSNs can consist of several underground nodes connected to a single aboveground hub (Liedmann and Wietfeld, 2017; Tiusanen, 2013) or a wide-area network combining multiple underground

nodes with multiple aboveground hubs (Froiz-Míguez et al., 2020).

Ritsema et al. (2009) were the first to monitor soil moisture in a golf course over several km$^2$ using a WUSN consisting of multiple locations. They used a complex array of 18 underground nodes, each installed at 0.1 m below soil surface, and a single aboveground hub. To bridge the distance between underground nodes and the single aboveground hub, they installed 24 additional aboveground nodes. Private engineering companies provided the network architecture (hardware and software), and

no further information was provided to allow reproducibility. More recently, Zhang et al. (2017) presented the Thoreau network as the first long-term WUSN for agricultural and environmental sensing. The network consists of a single hub installed at 41 m aboveground, which receives data from 27 underground nodes. Each node was buried at ~0.3 m while connected to soil moisture and water potential sensors. Using a wireless network named Sigfox (902 MHz radio band), they successfully covered an urban area of 2.5 km$^2$. Although some details on network architecture and power management were provided, there was no

detailed description of their customized hardware and software.

The extensive development of Internet of Things (IoT) hardware and IoT-related wireless communication protocols provides new opportunities for implementing communication solutions for WUSNs (Salam and Raza, 2020). García et al. (2020) and Vuran et al. (2018) present a comprehensive review of the most utilized wireless communication protocols, including Bluetooth, cellular, Wi-Fi, Sigfox, and ZigBee. Out of these, one of the most suitable for WUSN is the low-power long-range

(LoRa) network protocol (referred to here as LoRa-WUSN). LoRa is a relatively new, open technology designed for small data rates up to 50 kbps (Abrardo and Pozzebon, 2019; García et al., 2020) over aboveground distances of 10+ km assuming a clear line of sight (Sanchez-Iborra et al., 2018). LoRa-based networks have recently attracted increasing attention from academia and industry (Fraga-Lamas et al., 2019; Froiz-Míguez et al., 2020). Yet, most studies and implementations were done for aboveground LoRa networks, and LoRa-WUSN is considered an innovative field of research (Liedmann and Wietfeld,

2017) with scarce literature support to date.

Radio signal attenuation is a primary consideration for any type of WUSN, with the total attenuation strongly dependent on the length of the signal path in the soil (Bogena et al., 2009). According to Tiusanen (2013), there are four components affecting the signal attenuation between an underground node and an aboveground hub: signal loss due to soil medium attenuation, due to partial reflection from the soil surface, due to angular defocusing, and free air path loss. The first two components are

associated with the soil medium and the two latter with the air above. Signal quality, defined by the received signal strength index (RSSI; expressed using negative dBm units, closer to zero means greater signal strength) and signal-to-noise ratio (SNR;

positive values indicate more signal than noise), primarily depend on the transmitter operating frequency, burial depth, transmitter power, distance between the underground node and the aboveground hub, antenna type, data rate, soil moisture, and to a lesser extent on the soil texture and electrical conductivity (Bogena et al., 2009; Hardie and Hoyle, 2019). The limit of acceptable RSSI for wireless communication is subjective and determined based on experimental needs and the RSSI to SNR ratio (Hardie and Hoyle, 2019).

Alongside the advances made in IoT-related wireless communication, open-source hardware is an additional emerging field of interest in environmental research (Concialdi et al., 2020; Fisher and Gould, 2012; Froiz-Míguez et al., 2020). Open-source hardware consists of electronics that can be freely replicated or assembled using openly available instructions, such as schematics, drawings, and layouts (Chan et al., 2020). Arduino microcontrollers and Raspberry Pi microcomputers, with their software platforms, are perhaps the most common examples of open-source hardware. The widespread adoption of open-source hardware is led by hobbyists and the public and to a lesser extent by the academic community (e.g., the OPEnS Lab, Oregon State University). However, the available online information of tutorials, forums, and ongoing developments minimizes the learning curve and can help bridge the gap toward a higher implementation rate of open-source hardware in academic research (e.g., Levintal et al., 2020, 2021b; Reck et al., 2019; Weissman et al., 2020). Additional benefits of utilizing open-source hardware are: no prior experience with electronics or coding is needed though it can help (Chan et al., 2020), lower costs than existing commercial hardware (Levintal et al., 2021a), and the option for customized solutions. Specifically, harnessing open-source hardware for LoRa-WUSN lowers such sensor networks' cost and complexity, thus making them accessible for researchers.

The use of LoRa-WUSN in soil studies has not been comprehensively investigated (Hardie and Hoyle, 2019). The majority of studies on LoRa-WUSN can be found in engineering disciplines and focus on in-soil signal propagation (Wan et al., 2017) and antenna optimization (Salam and Raza, 2020), making it challenging to adapt their conclusions to other disciplines, such as hydrology and soil science. Moreover, there is a lack of studies showing the performance of LoRa-WUSN for long-term measurements (Cardell-Oliver et al., 2019). Most studies previously published on LoRa-WUSN are either proof-of-concept studies or short-lived laboratory experiments (e.g., Huang et al., 2020; Wan et al., 2017). In addition, to the best of our knowledge, there is only one study published to date that tested and presented results from LoRa-WUSN for soil measurements (Cardell-Oliver et al., 2019). None of the aforementioned studies, however, focused on providing step-by-step instructions for the design, assembly, and installation of WUSN by the end-user. Despite the rapid technological advancement of WUSNs, it seems the technology itself (assembly, programming) remains a major challenge to utilizing WUSNs more widely in environmental and academic research. By sharing detailed instructions on the design, assembly and installation of WUSNs, we think these systems can be more widely used by other scientists and adapted to individual research needs.

The aim of this study is to present a new open-source, low-cost, LoRa-WUSN system for measuring soil moisture and oxygen levels at multiple depths in an agricultural soil and to provide in detail the technical information for the system design,

assembly, programming, deployment, power management, and data analysis so that other researchers can adapt the system to their needs. Emphasis is given on modularity to allow the end-user to duplicate or change, if needed, and deploy without previous engineering knowledge. Therefore, hardware was limited to readily available components only. Eight sensors consisting of four digital soil moisture sensors and four analog oxygen sensors were connected to an underground open-source LoRa transmitter node, and an aboveground LoRa hub logged the received data. For validation, the system was deployed in a field planted with young poplar trees (*Populus trichocarpa*) for five months.

## 2 Materials and Methods

### 2.1 Hardware

Hardware was limited to readily purchasable products in order to assemble a low-cost LoRa-WUSN that can be easily replicated. The LoRa-WUSN installed in the field included two segments: a single underground node (to which the sensors were connected via underground cables) and a single aboveground hub combined with a datalogger (**Fig. 1**). **Table 1** summarizes the details of each sub-segment component. The core of the LoRa-WUSN is the Adafruit Feather M0 with RFM95 LoRa Radio (Adafruit, USA), hereafter called LoRa-Feather, which utilized a non-licensed 900 MHz radio band (or a 433 MHz in Europe). This is an open-source microcontroller with an embedded LoRa radio module, which is light and affordable (~$35). It also has multiple general-purpose input/output (GPIO) ports enabling connections to analog and digital sensors, and low power requirements (~0.7 mA standby, ~120 mA peak during 20 dBm transmission) (DeBell et al., 2019). The LoRa-Feather transmission power ranges between 5 and 20 dBm depending on the user choice; dBm (decibel-milliwatts) is the unit used for measuring transmission power output (Parri et al., 2019). We chose this model over other available LoRa-based microcontrollers because of three reasons: (1) the LoRa-Feather has a large set of free online tutorials and supporting information, making the development and integration relatively easy; (2) the LoRa-Feather was previously validated in aboveground LoRa-based experiments (DeBell et al., 2019); and (3) the Feather microcontroller has multiple extension boards (named FeatherWings) that can be mounted on the Feather, thus providing versatile capabilities, such as data logging, without additional hardware complexity.

The underground node (**Fig. 1c**) included a LoRa-Feather connected to an external omnidirectional antenna (900 Mhz Antenna Kit, Adafruit, USA) and a battery (Lithium-ion cylindrical battery - 3.7V 2200 mAh, Adafruit, USA). A power relay extension board was mounted on the LoRa-Feather to optimize the sensors' power consumption (Adafruit Non-Latching Mini Relay FeatherWing, Adafruit, USA). The relay provides power to four digital soil moisture sensors (5TM, METER Group, USA) and to an analog-to-digital converter (ADC) (ADS1015 12-Bit ADC, Adafruit, USA). The ADC was used to convert the data from four analog oxygen sensors (KE-25 Figaro Engineering Inc., Japan). We choose the Decagon 5TM combined soil moisture and temperature sensor and the Figaro KE-25 oxygen sensor due to their low-power requirements (5TM – 0.03 and 3 mA during sleep mode and measurement, respectively; KE-25 – no external power supply required for sensor operation),

low cost, and long-term use in soil monitoring (Oroza et al., 2018; Turcu et al., 2005; Weitzman and Kaye, 2017). All components, excluding the antenna and sensors, were placed in a waterproof enclosure and sealed using rubber coating (Performix 12213, Plasti Dip International, USA) to protect against expected soil water (**Fig. 1c**).

Underground nodes for WUSN need to be highly energy-efficient because the battery cannot be recharged without excavation (Hardie and Hoyle, 2019). To reduce the underground node's power consumption, which in our case measured and transmitted

sensor data every 1 or 2-hr, depending on scenario tested, we used two independent methods simultaneously. The first method was putting the LoRa-Feather into a low power consumption sleep mode. During the sleep mode, tested power requirements were reduced from ~40 mA during normal active mode or ~120 mA during peak transmission to ~0.035 mA, which translates to about seven years of LoRa-Feather operation using a 2200 mAh battery. However, this is a theoretical calculation because the sleep mode is deactivated during sensor measurements and data transmission and reception. The second method utilized

the power relay to eliminate the standby power draw from the four soil moisture sensors and the ADC (to which the four oxygen sensors were connected). The relay was closed (i.e., no power) during sleep mode and transmission/reception and turned on in each cycle for 5 s to allow sensor readings.

The aboveground hub (**Fig. 1b**) includes a LoRa-Feather connected to an external omnidirectional antenna (900Mhz Antenna Kit, Adafruit, USA) and a battery (Lithium-ion polymer battery - 3.7V 1200 mAh, Adafruit, USA). Received data were logged

on an SD card using an extension board mounted on the LoRa-Feather (Adalogger FeatherWing - RTC + SD, Adafruit, USA). The lithium battery can maintain only several days of continuous hub operation. Therefore, an external solar panel and a 12 V battery were connected to the built-in lithium battery charging module in the LoRa-Feather. Because the battery provided 12 V, a voltage converter to 5 V was used between the 12 V battery and the LoRa-Feather (UBEC DC/DC Step-Down Converter, Adafruit, USA). All components, excluding the antenna and solar panel, were placed in a waterproof enclosure (**Fig. 1b**).

The total cost of the system, apart from the sensors and solar panel, was $150 (**Table 1**). The sensors cost amounted to ~$1,050 for the four 5TM and four KE-25 sensors, yet this can vary depending on the number of sensors needed. In general, there is no limitation on the number of sensors connected to one underground node because the modular nature of open-source hardware allows the addition of hardware according to the user's needs. For example, adding four oxygen sensors can be achieved by adding a second ADC ($10, see **Table 1**) to the underground LoRa-Feather. However, more sensors will, of course, affect the

battery life.

Detailed assembly instructions for the underground node, sensors, and aboveground hub are provided in the Supplement Information (Section S1).

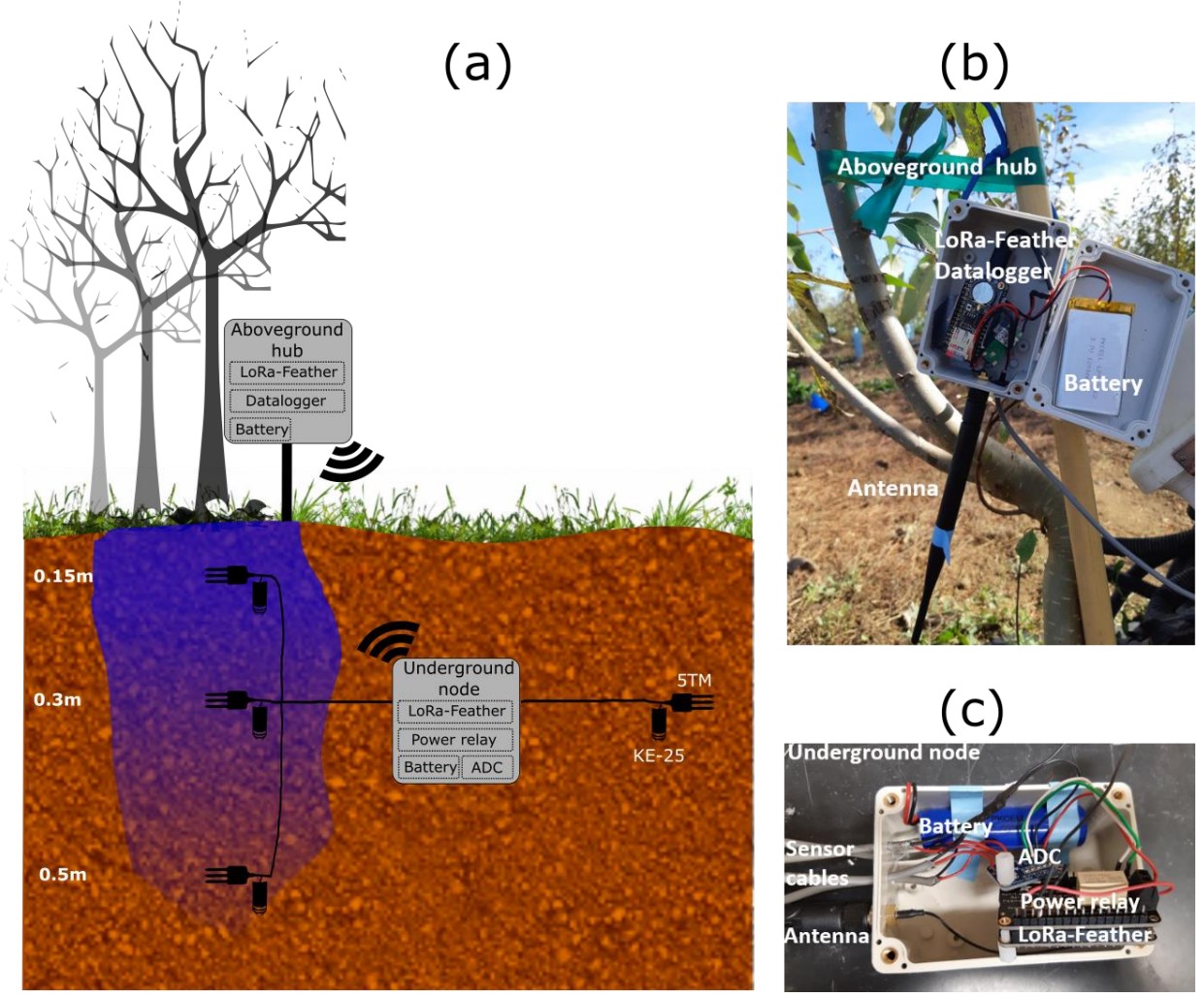

Figure 1: **Scheme of the LoRa-WUSN experimental setup in the field (a), the components of the aboveground hub (b), and the underground node before coating (c).**

Table 1: **Summary of hardware components and materials used in this study**.

| Component | Cost $ | Source of materials | Comments |
|---|---|---|---|
| **Underground node** | | | |
| Adafruit Feather M0 with RFM95 LoRa Radio - 900MHz – RadioFruit | 34.95 | Adafruit | Depending on the local regulation, use the 900MHz (e.g., USA) or the 433 MHz radio version (e.g., Europe) – both versions have similar capabilities and prices. |

| | | | |
|---|---|---|---|
| Header Kit for Feather - 12-pin and 16-pin Female Header Set | 0.95 | Adafruit | |
| 900Mhz Antenna Kit - For LoPy, LoRa, etc | 12.75 | Adafruit | |
| uFL SMT Antenna Connector | 0.75 | Adafruit | |
| Adafruit Non-Latching Mini Power Relay FeatherWing | 7.95 | Adafruit | |
| ADS1015 12-Bit ADC - 4 Channel with Programmable Gain Amplifier | 9.95 | Adafruit | For the four oxygen sensors |
| Short Feather Male Headers - 12-pin and 16-pin Male Header Set | 0.5 | Adafruit | |
| Lithium-Ion Cylindrical Battery - 3.7V 2200mAh | 9.95 | Adafruit | |
| LeMotech Waterproof Dustproof IP65 ABS Plastic Junction Box Universal Electric Project Enclosure Pale Gray 3.9 x 2.7 x 2 inch (100 x 68 x 50 mm) | 6.99 | Amazon | |
| **Sensors** | | | |
| KE-25 (oxygen sensor) | ~60 per sensor | Figaro | Four sensors were used ($240) |
| 50 mL Conical Centrifuge Tubes | ~1 per unit | Common lab equipment | Used to protect the KE-25 sensors |
| 5TM (soil moisture sensor) | ~200 per sensor | METER | Four sensors were used ($800) |
| Stereo Jack to Pigtail Probe Adapter, Brown and Orange | ~7 per unit | METER | For the 5TM sensors |
| **Aboveground hub** | | | |
| Adafruit Feather M0 with RFM95 LoRa Radio - 900MHz – RadioFruit | 34.95 | Adafruit | Depending on the local regulation, use the 900MHz (e.g., USA) or the 433 MHz radio version (e.g., Europe) – both versions are with similar capabilities and prices. |
| Header Kit for Feather - 12-pin and 16-pin Female Header Set | 0.95 | Adafruit | |
| 900Mhz Antenna Kit - For LoPy, LoRa, etc | 12.75 | Adafruit | |
| uFL SMT Antenna Connector | 0.75 | Adafruit | |
| Adalogger FeatherWing - RTC + SD Add-on For All Feather Boards | 8.95 | Adafruit | |
| CR1220 12mm Diameter - 3V Lithium Coin Cell Battery - CR1220 | 0.95 | Adafruit | For the RTC of the Adalogger |
| SD/MicroSD Memory Card (8 GB SDHC) | 9.95 | Adafruit | |
| Short Feather Male Headers - 12-pin and 16-pin Male Header Set | 0.5 | Adafruit | |

| | | | |
|---|---|---|---|
| Lithium-Ion Polymer Battery - 3.7V 1200mAh | 9.95 | Adafruit | |
| UBEC DC/DC Step-Down (Buck) Converter - 5V @ 3A output | 9.95 | Adafruit | |
| LeMotech Waterproof Dustproof IP65 ABS Plastic Junction Box Universal Electric Project Enclosure Pale Gray 3.9 x 2.7 x 2 inch (100 x 68 x 50 mm) | 6.99 | Amazon | |
| 12V waterproof solar panel | 59.95 | Amazon | The solar panel and battery used in this study were used for different experiments simultaneously, and they provided more power than was needed for the LoRa-WUSN. Cheaper options with instructions can be found on the Adafruit web page. |
| 12V battery | 18.99 | Amazon | |

## 2.2 Software

The LoRa-Feather microcontrollers were programmed using C++ in the commonly used open-source Arduino integrated
development environment (Chan et al., 2020). Existing Arduino compatible libraries were utilized and combined to configure and program the overall setup. The complete codes, with libraries and open license conditions, are described in the Supplement Information (section S2) and on Github at https://github.com/levintal/LoRaSystemForSoils. **Figs. 2a** and **2b** present the algorithm flow chart for the underground node and aboveground hub, respectively.

Every measurement cycle conducted by the underground node included 5 s of sensor readings followed by the transmission of
four data packets. Splitting the data into four packets was necessary because each packet is constrained to twenty chars (a char is a data type used in C++). The four packets included the measured data from the four oxygen and the four soil moisture sensors, and the node's measured battery voltage. An identifier value was assigned at the start of each data packet to mark its packet index (i.e., 1, 2, 3, or 4). After the transmission of the four data packets, the underground node waits for instructions from the aboveground hub on a new measurement interval or transmission power for the node's next measurement cycle. If
such a reply was received, then the node parameters were adjusted accordingly, e.g., increasing the next cycle transmission power from 5 to 20 dBm for cases in which stronger transmission power is needed. After each cycle, the underground node is set back to sleep mode.

The aboveground hub stays constantly in the receiver mode. At the end of a receiving cycle, once the four data packets from the underground node are received, the aboveground hub will send a reply to the underground node with instructions on the
new measurement interval or new transmission power, before data packets are written onto the SD card, together with the

RSSI, SNR, and a timestamp. If no reply was sent from the aboveground hub to the underground node, then the node will use its current measurement interval and transmission power for the next cycle.

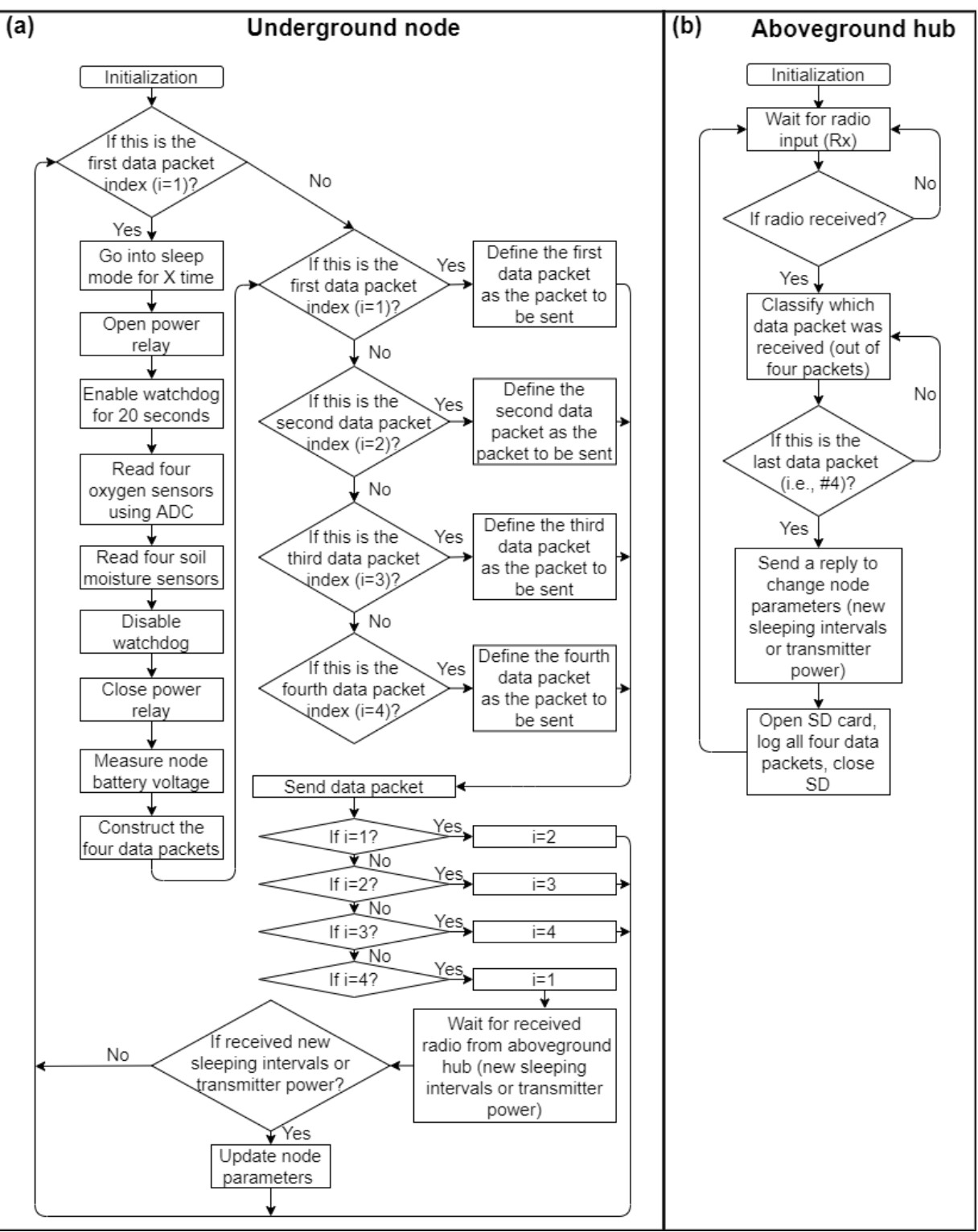

**Figure 2: Flow chart for underground node (a) and aboveground hub (b). A description for each step is given within the code (Supplement Information, Section S2).**

### 2.3 Field deployment

A field experiment was conducted to validate the proposed LoRa-WUSN (**Fig. 1a**). The system was installed in a poplar orchard irrigated with surface drip, located near Davis, CA, USA. The soil type at the site is a Reiff very fine sandy loam (SoilWeb), and the climate is Mediterranean with a total annual precipitation of about 500 mm and a mean annual temperature of 16.9 °C (Kourakos et al., 2019). The underground node was buried at 0.3 m below soil surface between two tree rows with the antenna in a horizontal orientation and pointed toward the aboveground hub, located on a nearby poplar tree at 1 m aboveground (**Fig. 1c**). The horizontal distance between the underground node and the aboveground hub was 2 m. The soil moisture and oxygen sensors were combined into four pairs. Three oxygen/soil moisture pairs were installed below one of the surface drip emitters at 0.15, 0.3, and 0.5 m, and the fourth pair at 0.3 m between the tree rows outside of the drip emitter's effective area. All sensors were connected to the underground node via underground cables buried at 0.3 m. The system was installed and tested throughout the winter season (November-2020 to March-2021). During this season, the soil has elevated soil moisture resulting from winter precipitation, which increases radio signal attenuation. Therefore, the wetter winter season, which is considered more challenging than the drier summer season when using LoRa-WUSN was chosen for this study. Atmospheric measurements were taken from meteorological station number 6 of the California Irrigation Management Information System (CIMIS), situated 1,300 m from the site.

To test the performance of the LoRa-WUSN, three experimental scenarios with different sleep modes and signal strengths during data transmission were tested in sequence between November 11, 2020 and March 31, 2021. The three scenarios include: (1) 2-hrs measurement intervals and low transmission power of 5 dBm (2-hrs, low transmission power) (11/10/2020-1/7/2021 and 2/27/2021-3/31/2021), (2) 1-hr measurement intervals and low transmission power of 5 dBm (1-hr, low transmission power) (1/8/2021-1/28/2021), and (3) 2-hrs measurement intervals and high transmission power of 20 dBm (2-hrs, high transmission power) (1/29/2021-2/25/2021). In addition to these three scenarios, we also tested a 1-min measurement interval and high transmission power of 20 dBm (2/26/2021) to assess the effect of the distance between the aboveground hub and the underground node on the wireless communication signal strength. In this scenario, the aboveground hub was positioned at different distances from the underground node ranging from 1, 10, 20, 30, to 50 m. At each distance, the aboveground hub was measuring for 10 min (1-min intervals) at a constant height of 2 m aboveground with the same antenna orientation. RSSI and SNR values from each location were used to assess communication strength. During the scenarios, the default LoRa-Feather parameters were used (bandwidth = 125 kHz, coding rate = 4/5, spreading factor = 128 chips/symbol, and CRC on) – additional information regarding these parameters can be found in the readme file link embedded within the code on Github.

## 3 Results and Discussion

Because the focus of this study is on the design and performance of an open-source LoRa-WUSN for measuring soil parameters, our results and discussion will mainly concentrate on the LoRa-WUSN's capabilities, such as battery and wireless communication performance, and not on the interpretation of the actual soil data collected in the field. The field data shown in **Fig. 3** is mainly used to highlight and validate performance aspects of the LoRa-WUSN and to stimulate possible future applications.

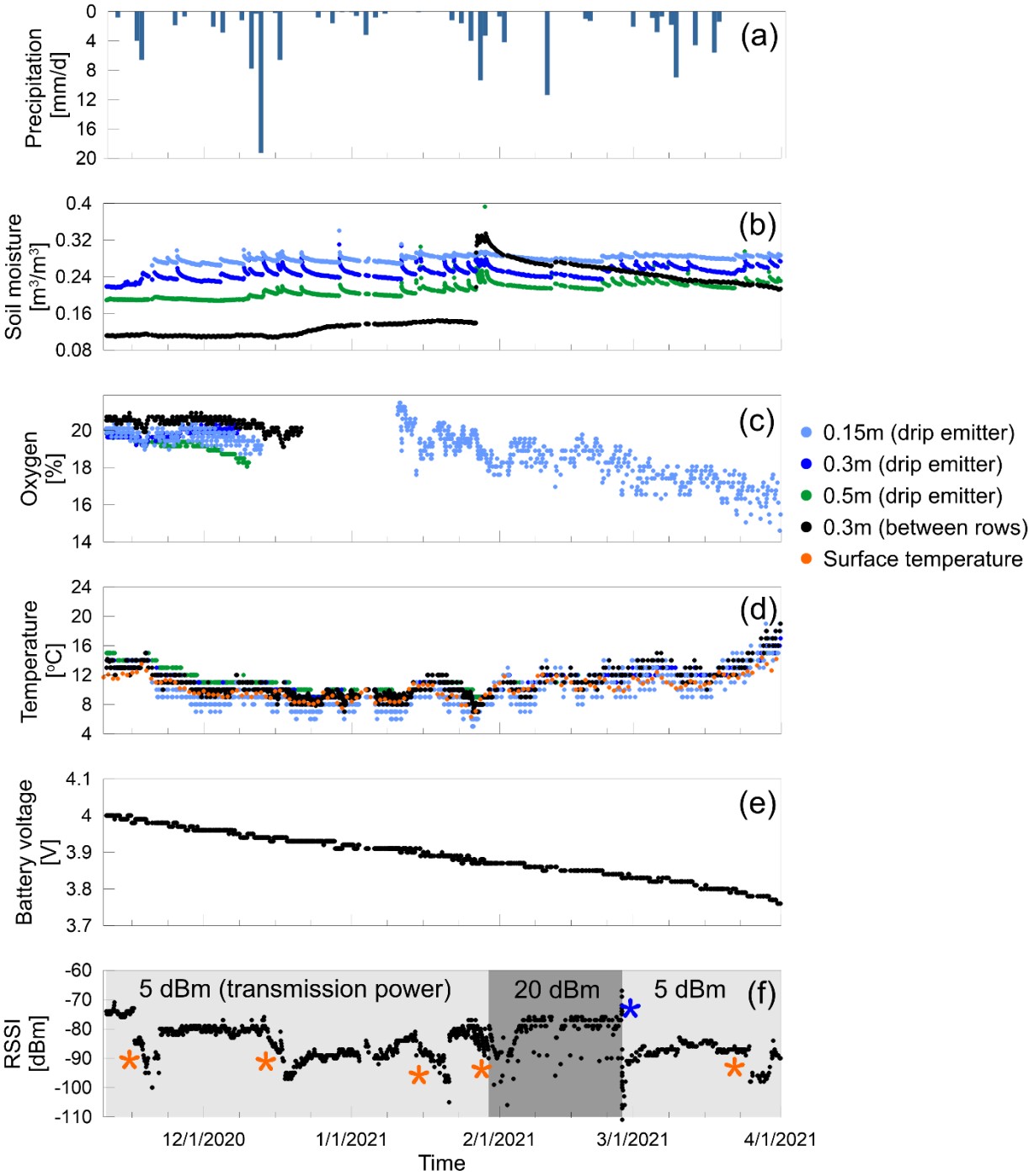


**Figure 3: Time series data from the field experiment. Orange asterisks in plot (f) represent the events during which the received signal strength index (RSSI) decreased markedly. The blue asterisk in plot (f) indicates the day of the distance**

**test (see Fig. 5). Blank areas in plot (c) are periods of missing data due to sensor malfunctions, as explained in section 3.1. The soil moisture sensor at 0.15 m stopped measuring after the initial field installation and was replaced two weeks**
**later.**

### 3.1 System performance

Soil moisture below the drip emitter (measured with the LoRa-WUSN) increased following each precipitation event and then, after the irrigation was restarted on February 22 2021 (**Fig. 3b,** color lines). Soil moisture between tree rows increased only after major precipitation events in mid-December and at the end of January (**Fig. 3b**, black line). Oxygen concentrations in the
soil were approximately 2-5 % lower than atmospheric concentrations with higher concentrations observed in the dry area between the tree rows than below the drip emitter (**Fig. 3c**). A general decrease trend in soil oxygen was observed during periods when soil moisture increased. All four oxygen sensors transmitted very low voltages (1-2 mV) corresponding to 0% soil oxygen content after several weeks of operation. This was likely due to clogging of the sensors' membranes, yet it was unexpected as we used a common sensor type (Kallestad et al., 2008; Turcu et al., 2005). The problem was solved by
embedding the sensor in a customized enclosure that contained an additional hydrophobic membrane (PTFE type). The added membrane blocked the water while still allowing gas exchange with the surrounding soil (see enclosure design in the Supplement Information, **Fig. S5**). We note that only the oxygen sensor at 0.15 m below the drip emitter was replaced with a new oxygen sensor and enclosure (1/10/2021) due to limited sensor availability; the clogged sensors were not reused due to the uncertainty regarding their performance and accuracy after clogging. Soil temperatures measured at the shallow depths
(0.15 and 0.30 m) showed a typical diel pattern as well as a seasonal trend of decreasing temperatures until the end of January (**Fig. 3d**). Temperatures were stable during February, followed by a sharp increase of ~4 °C during March at all measured depths.

The underground node's battery voltage was 4 V at the start of the field experiment and decreased to 3.77 V after five months of continuous operation (**Fig. 3e**). The battery decline rate was linear but varied depending on the measurement interval and
transmission power used. **Fig. 4** presents the battery decrease rates for the three main tested scenarios: 2-hrs between measurements with 5 dBm transmission power, 1-hr with 5 dBm transmission, and 2-hrs with 20 dBm transmission. Unexpectedly, the fastest voltage decrease rate was during the 2-hrs intervals with 5 dBm transmission (-0.0021 V/day), and not during the 1-hr intervals or higher transmission power of 20 dBm (-0.002 and -0.0012 V/day, respectively). This was most likely due to the increase in soil temperature at 0.3 m below soil surface during this measurement period (March), which was
3-4 °C higher than during the other scenarios (**Fig. 4**). In general, the amount of self-discharge of lithium-ion batteries is temperature-dependent with higher discharge rates observed at increased temperature.

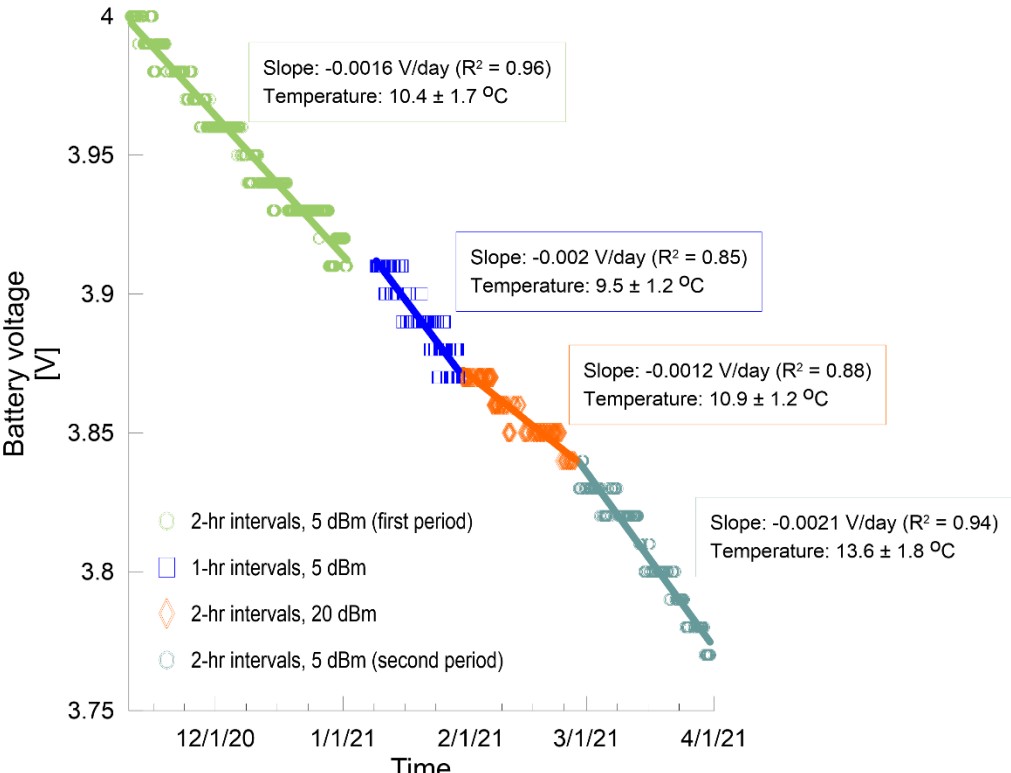

**Figure 4: Decrease rate of the underground node battery for three operational scenarios. The slopes were calculated using linear regressions (solid lines) and represent the average voltage decrease rate for each scenario. Temperature values represent the average soil temperature surrounding the node at 0.3 m below soil surface during that specific scenario.**

Fitting a linear regression to the battery voltage decrease rate allows estimating a total lifetime of the underground node's battery, assuming a 0.5 V range (i.e., from an initial 4.1 V charged battery to 3.6 V). The average battery decrease rate over the entire experiment was -0.0015 V/day ($R^2$ = 0.99), resulting in a battery life of ~333 days. We note that this estimation also includes the battery's self-discharge during sleep time under an average underground node temperature of 10.4 ± 1.8 °C, however, higher soil temperature will increase the battery's self-discharge rate, which usually ranges between 3-5% per month. Moreover, battery voltage decrease rate is not linear (Tarascon and Armand, 2001) and will be faster for a fully charged battery or once the battery is below the nominal voltage (~3.7 V). Therefore, the above battery life estimation is considered as the best-case scenario.

As power consumption is critical for WUSNs, we measured the duration and current of one complete operation cycle for four different transmission levels (5, 12, 17, and 20 dBm). The test was conducted in the lab using the same setting as in the field

with the addition of the INA260 (Adafruit, USA) to measure the current at 100-millisecond intervals. The results (**Table 2**, and **Fig. S10** and **S11** in the Supplement Information) indicate that the main parameter affecting power consumption is the duration of the sensor measurements (63 mA for 5 s), with a smaller contribution from the transmission (e.g., 129 mA for 0.3

s during 20 dBm transmission). We note that the power consumption during sleep mode was below the detection limit, and therefore, we used the value provided by the manufacturer of 0.035 mA. The end-user can use the power consumption values presented in **Table 2** to optimize system performance according to specific needs and batteries.

**Table 2: Power consumption**

| Stage | Average current [mA] | Average duration [s] |
|---|---|---|
| Sensor measurements | 63 | 5 |
| Transmission 5 dBm | 45 (60 peak) | 0.3 |
| Transmission 12 dBm | 78 (80 peak) | 0.3 |
| Transmission 17 dBm | 96 (109 peak) | 0.3 |
| Transmission 20 dBm | 129 (130 peak) | 0.3 |
| Receiver mode | 22 | 1 |
| Sleep mode | 0.035 | User-defined |

Two power management methods were used in the underground node, a power relay for the sensors (hardware type) and a sleep mode between measurement and transmission cycles (software type). If longer battery life is needed, there are additional power conservation methods available, such as cancelling a data transmission if measured values have not changed above a defined threshold compared to the previous measurement (Tiusanen, 2013), or reducing the number of data packets being sent by implementing an algorithm that reduces overall data size (Cardell-Oliver et al., 2019). Nevertheless, battery prices are

relatively low, and therefore, the best solution to extend battery life without complicating the system is to purchase (if needed) a battery with a larger capacity. In cases where extended battery life is needed, it is recommended to use battery technologies with lower self-discharge rates, such as non-rechargeable lithium-thionyl batteries with self-discharge rates lower than 1% per year. A comparison between different battery technologies is detailed in Callebaut et al. (2021). For instance, using a non-rechargeable lithium-thionyl battery with a ~7000 mAh is estimated to increase the underground battery's life threefold,

resulting in 2-3 years of operation (according to the power consumption presented in **Table 2**).

Average RSSI and SNR throughout the experiment were -84.4 ± 6 and 9.3 ± 0.6 dBm, respectively. Five continuous RSSI decrease events were identified, each lasting several days up to one week (**Fig. 3f**, orange asterisk symbols). Four out of the five events occurred 1-3 days after a major precipitation event (> 5 mm/d). However, the RSSI decrease observed in mid-January occurred after a week with zero precipitation. In general, no significant correlations (using linear regressions) were found between RSSI and precipitation or irrigation pulses or soil moisture at the different depths. Thus, we conclude that precipitation and soil moisture were not the only ambient conditions affecting signal strength. Zhang et al. (2017) came to a similar conclusion which they attributed to the environment's complexity. In other words, the real-world environment compared to lab conditions contains additional undetected parameters apart from soil moisture that reduce RSSI. Increasing transmission power from 5 to 20 dBm (1/29/2021-2/25/2021) improved the signal strength slightly, giving an average RSSI of -81.3 ± 5.7 dBm, which was higher than the average of the following month of March (-87.9 ± 3.5 dBm with low power transmission of 5 dBm). This can also be visually observed by the higher RSSI baseline during the high-power transmission scenarios (20 dBm) compared to the low-power transmission scenarios (5 dBm) (**Fig. 3f**).

The effective communication range between the underground node and aboveground hub was tested for two hours on 2/26/2021. This relatively short period was chosen to ensure similar ambient conditions throughout the test (similar relative humidity, temperature, soil moisture, etc.). The underground node was set to transmit at 1-min intervals and 20 dBm via a command from the aboveground hub. RSSI and SNR decreased with increasing distance between the aboveground hub and the underground node (**Fig. 5**). At the maximum distance tested (50 m), data packages from the underground node were still received and logged with an RSSI and SNR of -108.4 ± 1.7 dBm and 3.3 ± 1.8, respectively. The results agree with a LoRa-WUSN communication range test conducted by Hardie and Hoyle (2019) using an underground node at 0.3 m transmitting at 20 dBm and an aboveground hub. The authors tested LoRa RSSI and SNR results from four different soils (ranging from beach sand to clay loam) at distances ranging from 0 to 200 m. Their results show that even at 100 m, data packets were received by the aboveground hub, suggesting that similar to our setting, a distance greater than the 50 m tested in this study would be feasible if needed. Signal attenuation in the soil is an important parameter that will determine the maximum communication range. Bogena et al. (2009) provided a validated model that can be used to evaluate signal attenuation as a function of soil depth, soil moisture, and soil water electrical conductivity for different radio frequencies. A more detailed experimental analysis of in-soil LoRa signal range as a function of soil moisture and depth is presented by Wan et al. (2017). Different field settings may create additional complexity (Bogena et al., 2009), and there remains a need for further research in modelling and field validation of underground electromagnetic wave propagation, especially for clay soils in which soil moisture and bulk electrical conductivity are expected to be higher, thus reducing maximum communication range.

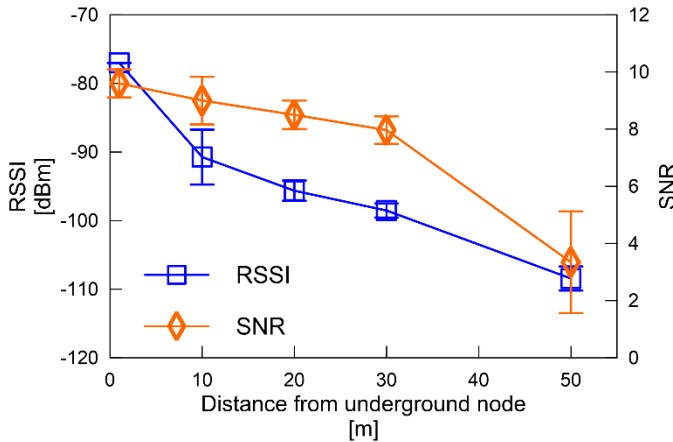


**Figure 5: Received signal strength index (RSSI) and signal-to-noise ratio (SNR) as a function of the linear distance between the underground node and the aboveground hub. The aboveground hub was positioned at five distances from the underground node (1, 10, 20, 30, and 50 m). At each distance, the hub recorded one reading per min for 10-min at a constant height of 2 m aboveground with the same antenna orientation. Transmission of the underground node was**

**set to 20 dBm. Error bars indicate the standard deviation from the average of 10 measurements at each distance.**

The chosen transmission power and radio band should also follow legal restrictions derived from local regulation. In Europe, for instance, the maximum approved transmission power is 14 dBm (for 433 MHz), compared to 30 dBm in the USA (915 MHz) (Fraga-Lamas et al., 2019; Froiz-Míguez et al., 2020; Haxhibeqiri et al., 2018). The results of our study show that even 5 dBm provided sufficient power for transmitting data from the underground node to the aboveground hub located at a

horizontal distance of 2 m (**Fig. 3**). The relationship between transmission power, underground node depth, distance between an underground node and an aboveground hub, and soil texture is discussed in Hardie and Hoyle (2019). We note that the authors used a radio band of 433 MHz compared to 915 MHz used in this study, and therefore, some differences are expected; lower radio band frequency will result in lower radio propagation losses (i.e., larger range) (Froiz-Míguez et al., 2020). To the best of our knowledge, there is no published comparison between the two radio bands for LoRa-WUSN, thus we cannot

conclude which of the two is preferable (e.g., in terms of RSSI or SNR). Another regulative limitation is the duty cycle for an on-air time. In Europe, it is 1 %, which means that for a 1 s LoRa transmission, this specific node cannot transmit during the following 99 s (Haxhibeqiri et al., 2018). The 0.3 s transmission duration presented in this study (**Table 2**) translates to a minimum interval time of 29.7 s before the subsequent transmission can be made.

Apart from RSSI and SNR, another critical parameter is the data packets receiving ratio, defined as the number of received

data packets at the aboveground hub divided by the packets that were sent from the underground node; a ratio of 100 % represents ideal conditions in which all sent packets were also received at first attempt. The average ratio during the experiment was 75 %, with higher ratio values observed at the start of the winter (~87 % during October) compared to the end (~50 %

during February). Lab test, conducted for five days using the same system setting (2-hr intervals, low power transmission of 5 dBm) resulted in a data packets receiving ratio of 100 %. By comparing the lab and field results, we conclude that the decrease

in received packets was due to electromagnetic interferences at the site. Potential sources of electromagnetic interferences were a nearby active airport situated 500 m to the south and an eddy-covariance flux tower situated at the same experimental site. Even with the low receiving ratio observed in February, the system was still able to transmit and store most of the data received from the underground node measured at the specified intervals. Missing data packets are an acceptable limitation for WUSN (Zhang et al., 2017). However, if needed, there are possible solutions to ensure higher data packet receiving ratios, as

discussed in the next section.

## 3.2 System modifications and configurations

In this study, we present a LoRa-WUSN that was built to measure soil moisture, soil temperature, and soil oxygen content. Nevertheless, the modular nature of open-source hardware allows the end-user various options for system configurations without adding substantial complexity. For example, instead of on-site data logging as conducted here, it is possible to add a

Wi-Fi component to the aboveground hub to get online real-time data (DeBell et al., 2019). If no Wi-Fi is in range, a cellular modem can replace the Wi-Fi component (Spinelli and Gottesman, 2019), which will be more costly due to the cellular service charges but it would provide greater flexibility and range. An alternative solution is sending the data from the aboveground hub at the experimental site to another aboveground LoRa station situated several kilometers from the site in an area with a Wi-Fi signal or Ethernet connectivity. Combined solutions, such as on-site data logging and Wi-Fi, Ethernet, or cellular

communication, are also possible (Levintal et al., 2021a).

Embedding a feedback mechanism within the code is possible if a packets receiving-ratio of 100 % is desired (i.e., all the sent data are also logged on the aboveground hub). The underground node continuously sends the same data packet until a reply from the aboveground hub is received stating that the packet was logged. The tradeoff of this modification is the increase in power consumption of the underground node due to the potentially greater number of transmission cycles. Power consumption

can be managed within the code by implementing a predefined threshold voltage below which the feedback mechanism will be disabled.

Installing multiple underground nodes at different locations is also feasible. This requires a simple software modification, in which every data packet (i.e., every singular transmission) is labeled at the start of the packet with another identifier specifying the underground node that sent the packet, and accordingly, the aboveground hub knows from which node the packet was

received. A similar method was presented by DeBell et al. (2019) for aboveground LoRa networks. We tested and validated this method in the lab using three nodes and a single hub. Using this approach simplifies system assembly for the end-user, however, it increases the risk for data packet loss in the cases of two nodes transmitting simultaneously. To quantify this risk, we conducted a test in which three nodes transmitted data packets at 1-min intervals for 20 hrs (i.e., 20 data packets per node).

Data packet receiving ratios were 100, 95, and 100 % for nodes 1, 2, and 3, respectively. These ratios indicate a low probability

for transmission collisions between nodes. Yet, if a significantly larger number of nodes is required, it is recommended to use more complex solutions like the LoRa Wide Area Networking technology (LoRaWAN). The LoRaWAN is an open-source protocol that uses the LoRa protocol to enable communication between multiple nodes and hubs (also referred to as gateways), with additional benefits such as adaptive data rates that can reduce power consumption (Froiz-Míguez et al., 2020; Haxhibeqiri et al., 2018). There is also an emerging use of LoRaWAN solutions commercialized by private companies. Yet, they are still

costly and, in most cases, target big end users, such as cities, and therefore, are less relevant for field-scale research. A review of the LoRaWAN technology is provided by Haxhibeqiri et al. (2018), and a more detailed focus on the limitations is provided by Adelantado et al. (2017).

## 4 Conclusions

This study presents a novel, low-cost wireless underground sensor network (WUSN) for soil monitoring using the relatively

new, open communication protocol named low-power long-range (LoRa). A field test, conducted for five months in an agricultural field, allowed assessing the system's capabilities. Soil moisture content, temperature, and soil oxygen concentrations were measured at three depths (0.15, 0.3, and 0.5 m) and data were transmitted from an underground node (0.3 m) to an aboveground receiving and logging hub. Communication tests showed an effective range of at least 50 m is possible between the underground node and aboveground hub. Using power management methods, battery life was estimated at ~333

days, with an option to triple this period when using a battery with a bigger capacity and lower self-discharge rate. The cost of all the data logging, power, and communication components was $150, one or two orders of magnitude smaller than other available commercial solutions. Emphasis was given on providing the complete technical guide and using only readily buyable hardware. By doing so, the technical and cost barriers were reduced, which we hope will allow easier reproducibility and open new applications for vadose zone and environmental monitoring studies.

**Data Availability**

The data is available within the above manuscript and the supplementary information.

**Author Contributions**

EL conceptualized and conducted the study and wrote the first manuscript draft. HED provided the resources. All the authors (EL, YG, GT, PFS, KS, and HED) contributed to the final version.

**Competing interests**

The authors declare that they have no conflict of interest.

**Acknowledgments**

This work was funded by the Gordon and Betty Moore Foundation and a Vaadia-BARD Postdoctoral Fellowship Award no. FI-605-2020. The authors would like to thank Cristina Prieto Garcia for her help with the field and laboratory experiments and

the two anonymous reviewers who helped improve this manuscript. This project was also supported by the USDA National Institute of Food and Agriculture, Hatch project number CA-D-LAW-2513-H, and the poplar field trial is funded as part of the DOE Genomics-Enabled Plant Biology for Determination of Gene Function, Office of Biological and Environmental Research award DE-SC0020164.

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
