# Peer review of "An underground, wireless, open-source, low-cost system for monitoring oxygen, temperature, and soil moisture"

_SOIL, 2021_

## Author Comment (AC1)

Referee #1

**General comments**

The authors present the design of an inexpensive underground soil sensor device that communicates wirelessly with an above-ground data logger, and that can be used to set up a network of wireless soil sensors. The strength of the study is that sensor device and datalogger are built with off-the-shelf components costing no more than 150 USD (without the soil sensors), and can be assembled by people with minimal electronics knowledge and without the need to design and construct a printed circuit board. All information needed to build it is given in the manuscript and the microprocessor scripts are made available on Github. In addition, the authors present results of a 5 months field testing during which the voltage of the battery of below-ground sensor device was monitored to assess how long such a device could operate without having to dig it up to recharge or replace the battery.

The very practical orientation of the manuscript is a strength, but at the same time a weakness in the sense that too little information is provided on power consumption as a function of measurement frequency, and about which performance can be expected in other soils that may have more radio signal attenuation hampering the wireless communication between the below-ground sensor device and the above-ground datalogger. The authors also did not measure power consumption, they only monitored battery voltage, and do not present information on the voltage -charge relationship for the battery they used. And some claims made about the autonomy of the device (up to 2-3 years with larger battery) and the possibility to scale it up to several sensor devices are not well founded.

The field test results they presented no doubt provided the authors with the necessary information to set up a wireless sensor network on their field site. But to make their work useful for colleagues who want to deploy it in different conditions, important information is missing. And that the above-ground datalogger is not capable of sending the data in real time to a server is a missed opportunity. Anybody who sets up a network of wireless soil sensors will also want to monitor the data in real time, and also that can be done with off the-shelf components.

The manuscript is written in clear language. It can be published in Soil and is relevant for the readership of Soil provided the authors provide extra information and justification for their claims, as explained in the specific comments below.

We want to thank the reviewer for the comments, which helped us improve the manuscript significantly. Detailed answers are given below using blue font. We would like to emphasize the main changes that we conducted:

(1) We added new results and related discussion on a power consumption test we conducted in response to the reviewer's comments. A discussion on other types of batteries was also added.

(2) A lab test was conducted to support the claim of connecting several nodes to one hub. In addition, we added a discussion on LoRaWAN as a more advanced solution.

(3) The discussion on radio signal attenuation was extended.

(4) We acknowledge that logging the data on an SD card without sending it in real-time is a disadvantage. References describing different real-time solutions are provided in the manuscript.

**Specific comments**

P1 L10: 'Wireless sensors pose the least disturbance to soil structure' needs to be formulated differently. The installation of a wireless sensor, i.e. the digging to install it, causes as much soil disturbance as for a wired sensor. The difference is that a wireless sensor that is installed below the tillage depth can stay in place for several years, while a wired sensor needs in many cases to be removed and installed again to allow field operations, in particular tillage, thus causing much more soil disturbance. That is the main selling point of wireless underground sensor devices, and that has to be made clear in the abstract, as well as in the first paragraph of the introduction.

We revised the phrasing in the abstract and the introduction.

"The use of wireless sensor networks to measure soil parameters eliminates the need to remove sensors for field operations, such as tillage, thus allowing long-term measurements without multiple disturbances to soil structure." (Abstract)

"A remaining challenge in vadose zone monitoring is the measurement of soil parameters, such as water content, without the need to remove sensors between field

operations, such as tillage, which often causes damage to wires connecting below-ground sensors to above-ground dataloggers." (Introduction)

P5 L129-136: two methods for reducing the power consumption of the below-ground sensor device are presented. But I guess the two methods (putting the Lora-Feather microprocessor in sleep mode and powering off the sensors) are complementary and were used simultaneously. That needs to be clarified.

We clarified the text.

"To reduce the underground node's power consumption, which in our case measured and transmitted sensor data every 1 or 2-hr, depending on scenario tested, we used two independent methods simultaneously."

P5 L131: power consumption of the LoRa-Feather during active mode and during sleep is given, and it is correctly explained that the fact the calculation that in sleep mode (35µA power consumption) the 2200 mAh battery can last 7 years is theoretical because most power is consumed when the module is doing and transmitting a measurement, and there is also the self-discharge by the battery. As power consumption is critical for an application where the sensor device needs an autonomy of several years (otherwise the advantage of an underground wireless device is largely lost), the authors need to report also the power consumption (in mAh, and best also duration of the active mode and its average power consumption in mA) during one measurement. That is essential to allow readers to assess what measurement frequency is possible for a given battery capacity.

We followed the reviewer's suggestion and conducted a power consumption test for each node cycle (i.e., sensor measurements, Tx, Rx, and sleep mode) for four different transmission levels. The results and discussion were added to the manuscript.

"As power consumption is critical for WUSNs, we measured the duration and current of one complete operation cycle for four different transmission levels (5, 12, 17, and 20 dBm). The test was conducted in the lab using the same setting as in the field with the addition of the INA260 (Adafruit, USA) to measure the current at 100-millisecond intervals. The results (**Table 2**, and **Fig. S10** and **S11** in the Supplement Information) indicate that the main parameter affecting power consumption is the duration of the sensor measurements (63 mA for 5 s), with a smaller contribution from the transmission

(e.g., 129 mA for 0.3 s during 20 dBm transmission). We note that the power consumption during sleep mode was below the detection limit, and therefore, we used the value provided by the manufacturer of 0.035 mA. The end-user can use the power consumption values presented in **Table 2** to optimize system performance according to specific needs and batteries.

**Table 2: Power consumption**

| Stage | Average current [mA] | Average duration [s] |
|---|---|---|
| Sensor measurements | 63 | 5 |
| Transmission 5 dBm | 45 (60 peak) | 0.3 |
| Transmission 12 dBm | 78 (80 peak) | 0.3 |
| Transmission 17 dBm | 96 (109 peak) | 0.3 |
| Transmission 20 dBm | 129 (130 peak) | 0.3 |
| Receiver mode | 22 | 1 |
| Sleep mode | 0.035 | User-defined |

"

Two related figures were also added in the Supplemental Information.

[Figure]

**Fig. S10**. Power consumption during one underground node cycle (sleep mode, sensor measurements, transmission, and receiver mode) for four different transmission levels (5, 12, 17, and 20 dBm).

[Figure]

**Fig. S11**. Power consumption during three consecutive underground node cycles (sleep mode, sensor measurements, transmission, and receiver mode) for four different transmission levels (5, 12, 17, and 20 dBm).

P8 L 169-171: An important aspect given the radio signal attenuation in soils is the power emitted by the antennas of the above-ground and below-ground devices. The authors report they used 5 and 23 dBm transmission power. The dipole antenna they used may have a gain of 2 dBi or so, so emitted power was probably +7 and +25 dBm. That is within the limits in the USA, Australia, India (+30 dBm), but too high for regulations in Europe (max is +14 dBm) or China (max is +12.15 dBm.) The authors need to point to the fact that there are such limits and discuss whether their system can still work in regions with such limits.

We added a new paragraph discussing power limitations due to local regulation. In addition, we added a reference describing the effect of transmission power on different operational parameters, such as distance between components and RSSI. We note that we mistakenly wrote 23 dBm as the maximum transmitting power in the first version. We corrected it throughout the manuscript to 20 dBm as the maximum transmitting power possible for this specific hardware (as stated in the manufacturer manual).

"The chosen transmission power and radio band should also follow legal restrictions derived from local regulation. In Europe, for instance, the maximum approved transmission power is 14 dBm (for 433 MHz), compared to 30 dBm in the USA (915 MHz) (Fraga-Lamas et al., 2019; Froiz-Míguez et al., 2020; Haxhibeqiri et al., 2018). The results of our study show that even 5 dBm provided sufficient power for transmitting data from the underground node to the aboveground hub located at a horizontal distance of 2 m (**Fig. 3**). The relationship between transmission power, underground node depth, distance between an underground node and an aboveground hub, and soil texture is discussed in Hardie and Hoyle (2019). We note that the authors used a radio band of 433 MHz compared to 915 MHz used in this study, and therefore, some differences are expected; lower radio band frequency will result in lower radio propagation losses (i.e., larger range) (Froiz-Míguez et al., 2020). To the best of our knowledge, there is no published comparison between the two radio bands for LoRa-WUSN, thus we cannot conclude which of the two is preferable (e.g., in terms of RSSI or SNR). Another regulative limitation is the duty cycle for an on-air time. In Europe, it is 1 %, which means that for a 1 s LoRa transmission, this specific node cannot transmit during the following 99 s (Haxhibeqiri et al., 2018). The 0.3 s transmission duration presented in this study (**Table 2**) translates to a minimum interval time of 29.7 s before the subsequent transmission can be made."

P15 L249-252: The authors calculate that the battery of their below-ground sensor device would last 333 days. But this calculation is only approximate as the relation between voltage and battery capacity is not linear. For the same reason, it is also dangerous to compare the slopes of the different scenario's in Fig. 4 to draw conclusions on discharge rates. The authors need to discuss this non-linearity at least.

We agree this is a simplified calculation. We added the additional factors and acknowledged the simplicity of the calculation in the text. In addition, we revised this paragraph and added the potential use of other types of batteries as suggested by the two reviewers.

"The average battery decrease rate over the entire experiment was -0.0015 V/day ($R^2 =$ 0.99), resulting in a battery life of ~333 days. We note that this estimation also includes the battery's self-discharge during sleep time under an average underground node temperature of 10.4 ± 1.8 °C, however, higher soil temperature will increase the battery's self-discharge rate, which usually ranges between 3-5% per month. Moreover, battery voltage decrease rate is not linear (Tarascon and Armand, 2001) and will be faster for a fully charged battery or once the battery is below the nominal voltage (~3.7 V). Therefore, the above battery life estimation is considered as the best-case scenario."

"In cases where extended battery life is needed, it is recommended to use battery technologies with lower self-discharge rates, such as non-rechargeable lithium-thionyl batteries with self-discharge rates lower than 1% per year. A comparison between different battery technologies is detailed in Callebaut et al. (2021). For instance, using a non-rechargeable lithium-thionyl battery with a ~7000 mAh is estimated to increase the underground battery's life threefold, resulting in 2-3 years of operation (according to the power consumption presented in **Table 2**)."

P15 L260: The authors explain that by simply increasing the capacity of their lithium-ion battery to 6000 mAh, they can increase the battery's life to 2-3 years. But that depends on how much the battery's self-discharge rate is. If that self-discharge rate is 5% per month, then that battery will not last 3 years, no matter how large the capacity. If it is only 2% per month, they can probably last 3 years. Unfortunately, LiPo battery manufacturers do not report self-discharge rates, and that rate is moreover dependent on temperature (it probably doubles for each temperature increase by 10K). So by 'extrapolating' from the 5-month field test that by increasing the battery capacity, it can

work for 2-3 years is quite uncertain. At least, the authors should discuss this aspect and point to the fact that non-rechargeable lithium-thionyl batteries exist which have a much lower self-discharge rate (1% per year) and are suited for IoT applications that need an autonomy of several years.

We revised this paragraph and added the suggestions as detailed in our answer to the previous comment.

P17 L231-234: The authors claim that additional underground nodes can be added at different locations, and that it only requires simple software modification. I doubt that this is true: if sensor devices are sending their payload at the same time (which undoubtedly will happen), how can the datalogger LoRa device handle that? This is normally solved in LoRa communication by using multichannel gateways. But the LoRa-Feather that the authors use for their above-ground datalogger node cannot work as a multichannel gateway. So the authors need to explain why they are sure that more nodes can simply be added (which is what you want to do to get a wireless sensor network). Also, what is then the practical distance between sensor nodes and data logger that is possible (the maximum distance)? In their discussion, the authors also need to explain to what extent their system is also going to work in other soil types. What if the dielectric permittivity of the soil is larger (higher water content) or the bulk electrical conductivity is higher (both conditions are likely in clay soils)? They may base this discussion on e.g. the paper by Bogena et al. (2009) that looked into these aspects (the paper is already cited in the manuscript under review)

We conducted an additional lab experiment in which three nodes transmitted data to one hub at 1-hr intervals for 20 hrs. The test showed a low ratio of data collisions (as described below), supporting our statement that adding more nodes is feasible using a relatively easy software change. We fully acknowledge that this solution cannot work if a larger number of nodes is needed (e.g., more than 20-30 nodes), and therefore, we added within the text a description of the LoRaWAN as an alternative solution. We emphasize that our objective was to reduce underground wireless systems' complexity to hopefully allow new users to explore this type of systems. Therefore, we are trying to keep system construction as simple as possible.

"Installing multiple underground nodes at different locations is also feasible. This requires a simple software modification, in which every data packet (i.e., every singular

transmission) is labeled at the start of the packet with another identifier specifying the underground node that sent the packet, and accordingly, the aboveground hub knows from which node the packet was received. A similar method was presented by DeBell et al. (2019) for aboveground LoRa networks. We tested and validated this method in the lab using three nodes and a single hub. Using this approach simplifies system assembly for the end-user, however, it increases the risk for data packet loss in the cases of two nodes transmitting simultaneously. To quantify this risk, we conducted a test in which three nodes transmitted data packets at 1-min intervals for 20 hrs (i.e., 20 data packets per node). Data packet receiving ratios were 100, 95, and 100 % for nodes 1, 2, and 3, respectively. These ratios indicate a low probability for transmission collisions between nodes. Yet, if a significantly larger number of nodes is required, it is recommended to use more complex solutions like the LoRa Wide Area Networking technology (LoRaWAN). The LoRaWAN is an open-source protocol that uses the LoRa protocol to enable communication between multiple nodes and hubs (also referred to as gateways), with additional benefits such as adaptive data rates that can reduce power consumption (Froiz-Míguez et al., 2020; Haxhibeqiri et al., 2018). There is also an emerging use of LoRaWAN solutions commercialized by private companies. Yet, they are still costly and, in most cases, target big end users, such as cities, and therefore, are less relevant for field-scale research. A review of the LoRaWAN technology is provided by Haxhibeqiri et al. (2018), and a more detailed focus on the limitations is provided by Adelantado et al. (2017)."

We did not test the maximum distance because our tested 50 m was sufficient for our needs. To address the lack of data, we expanded our discussion using the studies conducted by Bogena et al. (2009), Hardie and Hoyle (2019), and Wan et al. (2017).

"The results agree with a LoRa-WUSN communication range test conducted by Hardie and Hoyle (2019) using an underground node at 0.3 m transmitting at 20 dBm and an aboveground hub. The authors tested LoRa RSSI and SNR results from four different soils (ranging from beach sand to clay loam) at distances ranging from 0 to 200 m. Their results show that even at 100 m, data packets were received by the aboveground hub, suggesting that similar to our setting, a distance greater than the 50 m tested in this study would be feasible if needed. Signal attenuation in the soil is an important parameter that will determine the maximum communication range. Bogena et al. (2009) provided a validated model that can be used to evaluate signal attenuation as a function

of soil depth, soil moisture, and soil water electrical conductivity for different radio frequencies. A more detailed experimental analysis of in-soil LoRa signal range as a function of soil moisture and depth is presented by Wan et al. (2017). Different field settings may create additional complexity (Bogena et al., 2009), and there remains a need for further research in modelling and field validation of underground electromagnetic wave propagation, especially for clay soils in which soil moisture and bulk electrical conductivity are expected to be higher, thus reducing maximum communication range."

**Technical comments**

P2 L40: remove 'Zhang et al.,' from between the brackets of the reference.

Done.

P2 L48: 'Out of these,' instead of 'Out of which,'

Done.

Fig. 4: Y-axis title should read 'Battery voltage (V)'

Done.

P11 L196: 'March'

Done.

---

## Author Comment (AC2)

Referee #2

Environmental monitoring is changing towards distributed measurements over large areas using wireless sensor networks since many years. The next logical step is to extend this technology to underground operation. The main aspects are related to the electromagnetic wave propagation of the radio, power supply for long term operation (and water proofness).

The strong point of this paper is that it provides a guided path to a low cost underground wireless sensor network using readily available components. It is a good description and the open source nature of this project is highly appreciated. Therefore is a great starter for anyone interested in this field of research and encourages others to become part of this project. I guess this is one of the main goals of this paper, therefore a significant achievement and worth to be published.

We want to thank the reviewer for the comments, which we believe helped us improve the manuscript significantly. Detailed answers are given below using blue font.

There are two aspects which lack a bit:

1.) Underground wireless sensor networks already have entered the commercial domain. The first systems appeared on the market for farmers and for irrigation control in municipalities. In some parts of Europe LoRaWAN (and NB-IoT narrow band internet of things) networks are nationwide available even offering underground connectivity at some places. E.g. Czech IoT companies offer IP68 rated underground wireless soil moisture sensing for farming, golf courses and municipal parks. A short reference to such systems may be helpful.

We searched online but didn't find the references for the Czech IoT companies mentioned in the above comment. The only company providing fully buried wireless sensors we found is the Soil Scout (https://soilscout.com/solution/wireless-soil-moisture-sensor). Nevertheless, we added the suggested reference on the use of LoRaWAN networks by adding the following text to the manuscript.

"Yet, if a significantly larger number of nodes is required, it is recommended to use more complex solutions like the LoRa Wide Area Networking technology (LoRaWAN). The LoRaWAN is an open-source protocol that uses the LoRa protocol to enable communication between multiple nodes and hubs (also referred to as gateways), with additional benefits such as adaptive data rates

that can reduce power consumption (Froiz-Míguez et al., 2020; Haxhibeqiri et al., 2018). There is also an emerging use of LoRaWAN solutions commercialized by private companies. Yet, they are still costly and, in most cases, target big end users, such as cities, and therefore, are less relevant for field-scale research. A review of the LoRaWAN technology is provided by Haxhibeqiri et al. (2018), and a more detailed focus on the limitations is provided by Adelantado et al. (2017)."

2.) The authors may comment on using their proprietary LoRa radio protocol versus the standard LoRaWAN protocol. I understand that they wanted to optimized the protocol and it may be simpler to implement, but LoRaWAN has some good points as well like adaptive datarate and multichannel reception. Adaptive datarate can help to save power consumption when being close to the gateway. Multichannel could be beneficial in case of multipath propagation. The authors may also comment on the choice of the frequency. Usually lower frequencies are better for penetration soil. Besides 433 MHz instead of 900 MHz in some countries even a frequency around 170 MHz can be used which allows for a much larger range. As far as I understood the authors change only power level. What about changing LoRa spreading factor? Please also specify your LoRa settings. It may be looked up on Github, but would be good to see it in the paper.

We revised the paragraph and added the suggested references as detailed in our answer to the previous comment.

We did not change the spreading factor; however, following the reviewer's comment, we agree that this should be another parameter to test in future work. To clarify this, we added the LoRa settings to the Materials and Methods section:

"During all scenarios, the default LoRa-Feather parameters were used (bandwidth = 125 kHz, coding rate = 4/5, spreading factor = 128 chips/symbol, and CRC on) – additional information regarding these parameters can be found in the readme file link embedded within the code on Github."

Some further comments:

One reviewer mentioned that power regulations may vary between regions. In Europe there are also some frequencies in the 868 MHz range allocated for higher power (27dBm, but bandwidth and duty cycle dependent). So in order to comply with regulations are very thorough look into the frequency band plan is required, especially if higher power levels are requested.

We added a new paragraph discussing power limitations imposed by local regulation.

"The chosen transmission power and radio band should also follow legal restrictions derived from local regulation. In Europe, for instance, the maximum approved transmission power is 14 dBm (for 433 MHz), compared to 30 dBm in the USA (915 MHz) (Fraga-Lamas et al., 2019; Froiz-Míguez et al., 2020; Haxhibeqiri et al., 2018). The results of our study show that even 5 dBm provided sufficient power for transmitting data from the underground node to the aboveground hub located at a horizontal distance of 2 m (**Fig. 3**). The relationship between transmission power, underground node depth, distance between an underground node and an aboveground hub, and soil texture is discussed in Hardie and Hoyle (2019). We note that the authors used a radio band of 433 MHz compared to 915 MHz used in this study, and therefore, some differences are expected; lower radio band frequency will result in lower radio propagation losses (i.e., larger range) (Froiz-Míguez et al., 2020). To the best of our knowledge, there is no published comparison between the two radio bands for LoRa-WUSN, thus we cannot conclude which of the two is preferable (e.g., in terms of RSSI or SNR). Another regulative limitation is the duty cycle for an on-air time. In Europe, it is 1 %, which means that for a 1 s LoRa transmission, this specific node cannot transmit during the following 99 s (Haxhibeqiri et al., 2018). The 0.3 s transmission duration presented in this study (**Table 2**) translates to a minimum interval time of 29.7 s before the subsequent transmission can be made."

I am a bit confused with the power levels mentioned. In the summary of hardware components the RFM95 module is mentioned. According to the manufacturers datasheet a maximum output power of 20 dBm is possible. Do you really get 23 dBm out of the module?

Regarding the maximum transmitting power, we found a discrepancy between the datasheet and the example code provided by the manufacturer – in the datasheet, it is 20 dBm and in the code it is 23 dBm. This was our mistake and we would like to thank the reviewer for pointing this out. The maximum transmitting power was changed to 20 dBm throughout the manuscript, figures, and code.

As already stated by another reviewer battery technology is important. Lithium thionyl chloride batteries are state of the art with extremely low self discharge and relatively low costs. The reviewer is using D-cells with less than 1% self discharge per year and 19 Ah. Double D cells with 35 Ah are also available (size of underground enclosure is usually not critical). Energy

harvesting may be an option for unlimited lifetime but is still not mature enough and suffers from principle physical limits.

We revised this paragraph and added the potential use of other types of batteries as suggested by the two reviewers.

"In cases where extended battery life is needed, it is recommended to use battery technologies with lower self-discharge rates, such as non-rechargeable lithium-thionyl batteries with self-discharge rates lower than 1% per year. A comparison between different battery technologies is detailed in Callebaut et al. (2021). For instance, using a non-rechargeable lithium-thionyl battery with a ~7000 mAh is estimated to increase the underground battery's life threefold, resulting in 2-3 years of operation (according to the power consumption presented in **Table 2**)."

It may be a bit to deep in technical details, but please be careful when using the SDI-12 Arduino library with just the IO ports of the controller and not having an appropriate hardware interface according to the SDI-12 specification (see sdi-12.org). It may work, but may be out of specification.

We didn't encounter problems using SDI-12 Arduino library in our different systems. Nevertheless, we would like to thank the reviewer for providing these insights that we were not aware of. A link to this page was added to the code for cases where the end-user will want additional information related to SDI-12.

The authors discussed the range of their underground wireless system. I think there is still some demand for further research in modelling underground electromagnetic wave propagation, probably not within this paper but in future research.

We added this suggestion to the discussion.

"Different field settings may create additional complexity (Bogena et al., 2009), and there remains a need for further research in modelling and field validation of underground electromagnetic wave propagation, especially for clay soils in which soil moisture and bulk electrical conductivity are expected to be higher, thus reducing maximum communication range."